**Peer Review** The peer review history for this article is available as a PDF in the Supporting Information.

**Key Points:**

- Space-ground conjugate observations point to magnetospheric whistler-mode waves as the driver of ionospheric TEC perturbations (dTEC)
- The amplitude spectra of dTEC and whistlers are consistent and the cross-correlation between modeled and observed dTEC reaches 0.8
- Whistler-mode wave amplitudes and dTEC are modulated by ULF waves, which exhibit concurrent compressional and poloidal mode variations

**Supporting Information:**

Supporting Information may be found in the online version of this article.

**Correspondence to:**

Y. Shen,
yshen@epss.ucla.edu

**Citation:**

Shen, Y., Verkhoglyadova, O. P., Artemyev, A., Hartinger, M. D., Angelopoulos, V., Shi, X., & Zou, Y. (2024). Magnetospheric control of ionospheric TEC perturbations via whistler-mode and ULF waves. *AGU Advances*, 5, e2024AV001302. https://doi.org/10.1029/2024AV001302

**Author Contributions:**

**Conceptualization:** Yangyang Shen, Anton Artemyev, Michael D. Hartinger
**Data curation:** Olga P. Verkhoglyadova, Xueling Shi
**Formal analysis:** Yangyang Shen, Olga P. Verkhoglyadova, Anton Artemyev, Michael D. Hartinger, Vassilis Angelopoulos, Xueling Shi, Ying Zou
**Investigation:** Yangyang Shen, Olga P. Verkhoglyadova, Anton Artemyev, Michael D. Hartinger,

# Magnetospheric Control of Ionospheric TEC Perturbations via Whistler-Mode and ULF Waves

Yangyang Shen[1] ![ORCID], Olga P. Verkhoglyadova[2] ![ORCID], Anton Artemyev[1] ![ORCID], Michael D. Hartinger[1,3] ![ORCID], Vassilis Angelopoulos[1] ![ORCID], Xueling Shi[4] ![ORCID], and Ying Zou[5]

[1]Department of Earth, Planetary, and Space Sciences, University of California, Los Angeles, CA, USA, [2]Jet Propulsion Laboratory, California Institute of Technology, Pasadena, CA, USA, [3]Space Science Institute, Center for Space Plasma Physics, Boulder, CO, USA, [4]Department of Electrical and Computer Engineering, Virginia Tech, Blacksburg, VA, USA, [5]Johns Hopkins University Applied Physics Laboratory, Laurel, MD, USA

**Abstract** The weakly ionized plasma in the Earth's ionosphere is controlled by a complex interplay between solar and magnetospheric inputs from above, atmospheric processes from below, and plasma electrodynamics from within. This interaction results in ionosphere structuring and variability that pose major challenges for accurate ionosphere prediction for global navigation satellite system (GNSS) related applications and space weather research. The ionospheric structuring and variability are often probed using the total electron content (TEC) and its relative perturbations (dTEC). Among dTEC variations observed at high latitudes, a unique modulation pattern has been linked to magnetospheric ultra-low-frequency (ULF) waves, yet its underlying mechanisms remain unclear. Here using magnetically conjugate observations from the THEMIS spacecraft and a ground-based GPS receiver at Fairbanks, Alaska, we provide direct evidence that these dTEC modulations are driven by magnetospheric electron precipitation induced by ULF-modulated whistler-mode waves. We observed peak-to-peak dTEC amplitudes reaching $\sim 0.5$ TECU (1 TECU is equal to $10^6$ electrons/m$^2$) with modulations spanning scales of $\sim 5$–100 km. The cross-correlation between our modeled and observed dTEC reached $\sim 0.8$ during the conjugacy period but decreased outside of it. The spectra of whistler-mode waves and dTEC also matched closely at ULF frequencies during the conjugacy period but diverged outside of it. Our findings elucidate the high-latitude dTEC generation from magnetospheric wave-induced precipitation, addressing a significant gap in current physics-based dTEC modeling. Theses results thus improve ionospheric dTEC prediction and enhance our understanding of magnetosphere-ionosphere coupling via ULF waves.

**Plain Language Summary** Radio signals are refracted or diffracted as they traverse the ionosphere filled with free electrons. The ionosphere TEC, which is the total number of electrons along the raypath from the satellite to a receiver, helps to correct refractive errors in the signal, while its relative perturbations dTEC can be used to probe diffractive fluctuations known as ionosphere scintillation. Refractive error degrades GNSS positioning service accuracy while scintillation leads to signal reception failures and disrupts navigation and communication. Thus, an accurate understanding and modeling of TEC and dTEC is vital for space weather monitoring and GNSS-related applications. This study analyzes conjugate observations of ionospheric dTEC from a ground-based GPS receiver and magnetospheric whistler-mode waves (a distinct type of very-low-frequency electromagnetic waves) from the THEMIS spacecraft, which were well-aligned both in time and space. We find a good cross-correlation ($\sim 0.8$) between observed and modeled dTEC, driven by whistler-induced magnetospheric electron precipitation. These results point to whistler-mode waves as the driver of the observed dTEC. Both dTEC and whistler-mode wave amplitudes were modulated by ULF waves. These findings enhance physics-based ionospheric TEC prediction and our understanding of magnetosphere-ionosphere coupling.

## 1. Introduction

The Earth's ionosphere contains weakly ionized plasma in the atmosphere between approximately 80 and 1,000 km altitude. The state of ionospheric plasma is controlled by a complex interplay between solar and magnetospheric inputs from above, neutral atmospheric processes from below, and plasma electrodynamics from within. The resulting structuring and variability of ionospheric plasma have a major, adverse impact on the global navigation satellite system (GNSS) radio signals as they propagate through the ionosphere and experience varying degrees of refraction and diffraction (Morton et al., 2020). Refraction causes signal group delay and phase

Vassilis Angelopoulos, Xueling Shi,
Ying Zou
**Methodology:** Yangyang Shen, Olga
P. Verkhoglyadova, Anton Artemyev,
Michael D. Hartinger, Xueling Shi,
Ying Zou
**Resources:** Michael D. Hartinger,
Vassilis Angelopoulos
**Software:** Olga P. Verkhoglyadova,
Vassilis Angelopoulos, Xueling Shi
**Supervision:** Anton Artemyev,
Vassilis Angelopoulos
**Validation:** Yangyang Shen, Olga
P. Verkhoglyadova, Anton Artemyev,
Michael D. Hartinger, Xueling Shi
**Writing – original draft:** Yangyang Shen
**Writing – review & editing:**
Yangyang Shen, Olga P. Verkhoglyadova,
Anton Artemyev, Michael D. Hartinger,
Vassilis Angelopoulos, Xueling Shi,
Ying Zou

advance, leading to dominant errors in GNSS position, velocity, and time solutions, while diffraction causes stochastic intensity and phase fluctuations at the receiver, commonly known as ionospheric scintillation (Rino, 2011; Yeh & Liu, 1982). Scintillation leads to increased GNSS receiver measurement noise and errors and, in extreme cases, phase-tracking loss of lock or signal reception failures (Kintner et al., 2007). Thus, these ionospheric effects pose real threats to the reliability, continuity, and accuracy of GNSS operations and applications (Coster & Yizengaw, 2021; Morton et al., 2020). Understanding the causes for ionospheric structuring and variability is critical for forecasting their impacts on GNSS applications—a long-standing challenge for space weather research (Hey et al., 1946; Jakowski et al., 2011; Morton et al., 2020). The importance of this ionosphere forecasting has recently gained increased attention as the solar maximum unfolds and concerns over space weather events such as geomagnetic storms loom large (e.g., Hapgood et al., 2022; Kintner et al., 2007; Pulkkinen et al., 2017).

Ionospheric refraction is typically quantified by the total electron content (TEC), which is the total number of electrons within a unit cross section along the raypath extending from the receiver to the satellite. For dual-frequency GNSS or Global Positioning System (GPS) receivers, the TEC is estimated from differential group delays and carrier-phase advances (Ciraolo et al., 2007; Mannucci et al., 1998; McCaffrey & Jayachandran, 2017). Global TEC maps, constructed from networks of GNSS receivers on the ground and in orbit, can be used not only to correct ionospheric effects in GNSS-related applications but also to monitor large- and meso-scale traveling ionospheric disturbances, typically exceeding 100 km in horizontal wavelength (Hunsucker, 1982; Themens et al., 2022; S.-R. Zhang et al., 2022). Traveling ionospheric disturbances may result from internal ionospheric dynamics or from atmospheric effects from below linked to natural hazards, such as tsunamis, earthquakes, explosions, and volcanic eruptions (Astafyeva, 2019; Komjathy et al., 2016). High-resolution TEC from individual receivers and its relative perturbations dTEC and rate of changes (ROTI) are often used for detecting small-scale ionospheric irregularities and scintillation events (Cherniak et al., 2014; Makarevich et al., 2021; McCaffrey & Jayachandran, 2019; Nishimura et al., 2023; Pi et al., 1997).

While empirical and climatological TEC models exist (Jakowski et al., 2011; Rideout & Coster, 2006), physics-based modeling of TEC perturbations remains challenging. One of the main challenges in physical modeling of dTEC and space weather prediction is the complex structuring and variability of ionosphere plasma. Rapid (<a few minutes) and small-scale (<∼100 km) dTEC are observed at both low and high latitudes but generated by distinct mechanisms and drivers (Basu et al., 2002; Fæhn Follestad et al., 2020; Jin et al., 2015; Kintner et al., 2007; Moen et al., 2013; Pi et al., 1997; Pilipenko et al., 2014; Prikryl et al., 2015; Spogli et al., 2009; Watson, Jayachandran, Singer, et al., 2016). Near equatorial latitudes, these small-scale dTEC result from plasma bubbles or density depletions formed around post-sunset, primarily driven by the Rayleigh-Taylor instability associated with lower atmosphere-ionosphere coupling processes (C.-S. Huang & Kelley, 1996; Kelley, 2009; Xiong et al., 2010; Aa et al., 2020; Jin et al., 2020). At high latitudes, dTEC are associated with plasma irregularities in the auroral, cusp, and polar cap regions, spanning a few meters to hundreds of kilometers in spatial scale (e.g., Basu et al., 1990; Moen et al., 2013; Spicher et al., 2017). These irregularities are primarily driven by solar-magnetosphere-ionosphere coupling, which involves a complex interplay and synergy among solar extreme-ultraviolet radiation, plasma $\vec{E} \times \vec{B}$ drifts, charged-particle precipitation into the atmosphere, magnetic field-aligned currents, and various ionospheric plasma instabilities (Fæhn Follestad et al., 2020; Kelley, 2009; Moen et al., 2013; Spicher et al., 2015).

Among dTEC variations observed near the auroral latitudes, a unique modulation pattern has been linked to magnetospheric ultralow frequency (ULF) waves (Davies & Hartmann, 1976; Okuzawa & Davies, 1981; Pilipenko et al., 2014; Skone, 2009; Watson et al., 2015; Watson, Jayachandran, Singer, et al., 2016; Zhai et al., 2021). These ULF waves feature broadband or quasi-monochromatic geomagnetic pulsations with periods from about 0.2 to 600 s (Jacobs et al., 1964) and are considered to be crucial for energy and plasma transport throughout the solar-magnetosphere-ionosphere-thermosphere system (e.g., Southwood & Kivelson, 1981; M. K. Hudson et al., 2000, 2008; Hartinger et al., 2015, 2022; Zong et al., 2017). Skone (2009) noted that average power of ground-based ULF waves and dTEC exhibited similar temporal variations in the Pc3 band (∼22–100 mHz). Pilipenko et al. (2014) observed a high coherence (∼0.9) between dTEC and global Pc5 pulsations in a few mHz during a geomagnetic storm. Watson, Jayachandran, Singer, et al. (2016) also reported a high coherence and common power between dTEC and ULF radial magnetic field variations in the Pc4 band (6.7–22 mHz). Fully understanding ULF-induced ionospheric dTEC not only enhances the ionosphere forecasting during space

weather events but also elucidates the critical pathways of geospace energy coupling and dissipation via ULF waves.

To date, despite numerous proposals for direct dTEC modulation mechanisms by ULF waves (Pilipenko et al., 2014), no mechanism has yet been conclusively established. Recently, Wang et al. (2020) have reported a storm-time event where duskside ionospheric density was modulated by ULF waves in the Pc5 range. Pc5-modulated density variations observed from radar data were used to infer modulated precipitating electrons over an energy range of ∼1–500 keV and an altitude range of ∼80–200 km. Higher-energy precipitating electrons deposit their energy and induce impact ionization at lower altitudes, whereas lower-energy electrons do so at higher altitudes. The authors postulated that the precipitation and density perturbations are likely due to electron pitch-angle scattered into the loss cone by ULF-modulated very low frequency whistler-mode waves.

This postulation of whistler-driven dTEC is supported by extensive observations and models that demonstrate that ULF waves often coexist with and modulate whistler-mode waves (Coroniti & Kennel, 1970; W. Li, Thorne, et al., 2011; W. Li, Bortnik, Thorne, Nishimura, et al., 2011; Watt et al., 2011; Jaynes et al., 2015; Xia et al., 2016, 2020; X.-J. Zhang et al., 2019; X. J. Zhang et al., 2020; X. Shi et al., 2022; L. Li et al., 2022, 2023). The modulation of the whistler-mode wave growth is potentially attributed to compression-induced ambient thermal or resonant hot electron density variations (W. Li, Bortnik, Thorne, Nishimura, et al., 2011; Xia et al., 2016, 2020; X.-J. Zhang et al., 2019; X. J. Zhang et al., 2020), resonant electron anisotropy variations (W. Li, Thorne, et al., 2011; Watt et al., 2011), and nonlinear resonant effects from periodic magnetic field configuration variations (L. Li et al., 2022; L. Li et al., 2023). The periodic excitation of whistler-mode waves at ULF wave frequencies leads to periodic electron precipitation, which drives pulsating auroras (e.g., Jaynes et al., 2015; Miyoshi et al., 2010; Nishimura et al., 2010) and potentially explains many previously reported dTEC modulations at ULF frequencies (Pilipenko et al., 2014; Watson, Jayachandran, Singer, et al., 2016; Zhai et al., 2021).

However, it is challenging to establish a direct link between magnetospheric drivers and ionospheric dTEC during ULF modulation events due to several complicating factors: (a) the path-integrated nature of dTEC, which strongly depend on the satellite-to-receiver raypath elevation (e.g., Jakowski et al., 1996; Komjathy, 1997), (b) inherent phase shifts due to coexisting propagation and modulation effects (Watson et al., 2015), particularly when conjugate observations are misaligned or not synchronized, and (c) the dynamic and turbulent nature of the auroral ionosphere (Kelley, 2009). Direct evidence linking dTEC to magnetospheric drivers is yet to be identified.

In this study, conjugate observations from the THEMIS spacecraft and the GPS receiver at Fairbanks, Alaska (FAIR) allow us to identify the driver of GPS dTEC as magnetospheric electron precipitation induced by ULF-modulated whistler-mode waves. Figure 1 illustrates the physical picture emerging from these magnetically conjugate magnetospheric and ionospheric observations of ULF waves, modulated whistler-mode waves, electron precipitation, and dTEC.

In what follows, Section 2 describes data sets and models employed to estimate whistler-driven precipitation and resulting dTEC. Section 3 presents a detailed analysis and cross-correlation between observed and modeled dTEC. Section 4 discusses the geophysical implications and applications of our results, which are followed by the main conclusions.

## 2. Data and Methodology

We derive 1-s TEC measurements from phase and pseudorange data collected by the GPS receiver at FAIR during 15:06–16:36 UT on 3 July 2013, processed at the Jet Propulsion Laboratory using the GipsyX and Global Ionospheric Mapping software (Bertiger et al., 2020; Komjathy et al., 2005). Phase-based TEC measurements are leveled using pseudorange delays for each phase-connected data collection. We focus on links between FAIR and GPS satellites with pseudo random noise numbers 40, 43, and 60, referred to as GPS40, GPS43, and GPS60, whose ionospheric pierce points at 300 km altitude are within 200 km proximity to FAIR, or pierce points at 150 km within 100 km proximity to FAIR, to ensure relatively high elevation angles and thus better observation geometry to resolve dTEC.

The pierce point of 300 km altitude is selected based on the measured F2-region peak density height hmF2 from the ground-based ionosonde located at the Eielson station (64.66°N, 212.03°E) in Supporting Information S1. While the background density peaks at ∼300 km in the F2 region, the modulation of dTEC may be located at lower altitudes. Thus, we also present results using an ionosphere pierce point at 150 km altitude. The obtained

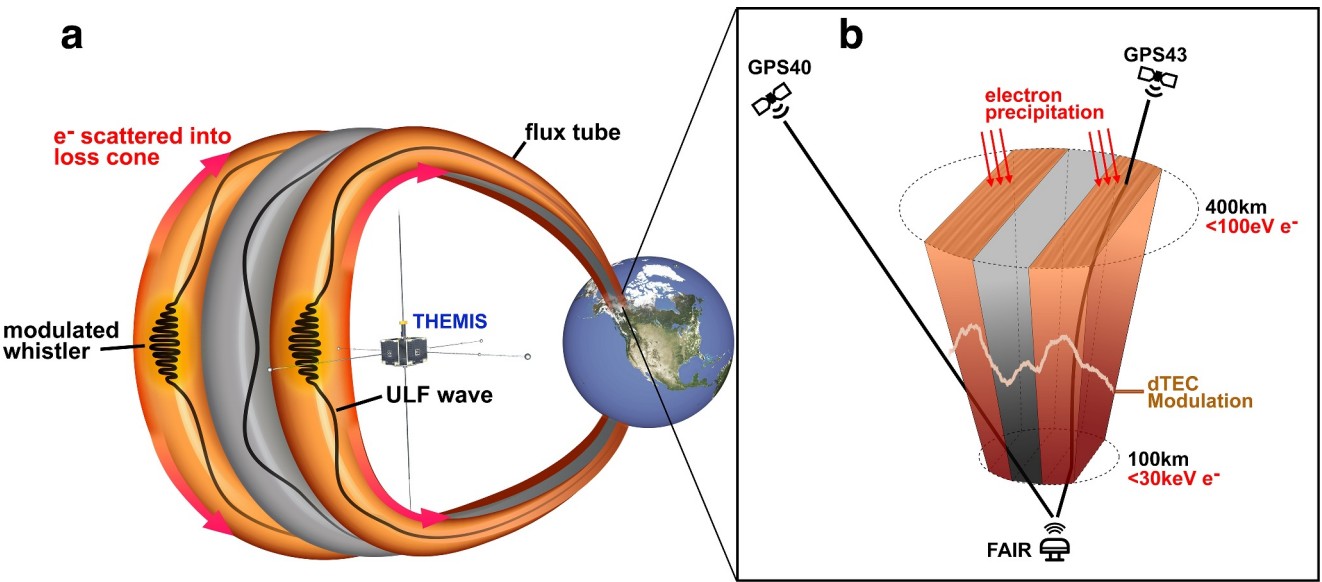

**Figure 1.** Schematic diagram showing coordinated observations from THEMIS and FAIR of (a) modulation of whistler-mode waves near the magnetic equator by ULF waves, electron pitch-angle scattering into the loss cone, and precipitation into the ionosphere (red arrows) induced by modulated whistler-mode waves; and (b) the modulated electron precipitation with energies of ∼0.1–30 keV deposits their energies at altitudes between ∼100–400 km and induces modulated impact ionization and dTEC having amplitudes as large as ∼0.5 TECU and spanning scales of ∼5–100 km. This dTEC modulation was captured by the signal from GPS43, which has a high elevation, but was overlooked by the signal from GPS40, which has a relatively lower elevation.

TEC is expressed in TEC units (TECU), that is, $10^{16}$ electrons/m$^2$. The slant TEC is converted to VTEC using the standard mapping function (e.g., Mannucci et al., 1998). Measurements with elevation angles less than 30° are excluded to reduce multipath effects (Jakowski et al., 1996). The VTEC data are then detrended to get dTEC using a fourth-order Butterworth lowpass filter. The low pass filter has a cutoff period of 25 min, to focus on ULF-related perturbations and reduce contributions from medium- and large-scale traveling ionosphere disturbances (Hunsucker, 1982).

We use the following data sets from THEMIS E (Angelopoulos, 2008): electron energy and pitch-angle distributions measured by the Electrostatic Analyzers instrument in the energy range of several eV up to 30 keV (McFadden et al., 2008), DC vector magnetic field at spin resolution (∼3 s) measured by the Fluxgate Magnetometers(Auster et al., 2008), electric and magnetic field wave spectra within 1 Hz–4 kHz, measured every ∼8 s by the Digital Fields Board, the Electric Field Instrument, and the search coil magnetometer (Bonnell et al., 2008; Cully, Ergun, et al., 2008; Le Contel et al., 2008). Background electron densities are inferred from spacecraft potentials (Bonnell et al., 2008; Nishimura et al., 2013). We also use ground-based magnetometer measurements every 1 s from the College (CMO) site operated by the United States Geological Survey Geomagnetism Program and from the Fort Yukon (FYKN) site operated by the Geophysical Institute at the University of Alaska.

THEMIS observations of electron distributions and wave spectra allow us to calculate the precipitating flux of electrons scattered into the loss cone by whistler-mode waves using quasilinear diffusion theory (Kennel & Engelmann, 1966; Lyons, 1974). For whistler-mode wave normals $\theta < 45°$, we use a validated analytical formula of bounce-averaged electron diffusion coefficients from Artemyev et al. (2013). For small pitch angle $\alpha_{eq}$ approaching the loss cone $\alpha_{LC}$, the first-order cyclotron resonance provides the main contribution to the bounce-averaged diffusion rate:

$$\langle D_{\alpha_{eq}\alpha_{eq}} \rangle \simeq \frac{\pi B_w^2 \Omega_{ceq} \omega_m}{4\gamma B_{eq}^2 \Delta\omega (p\epsilon_{meq})^{13/9} T(\alpha_{LC}) \cos^2_{\alpha_{LC}}} \times \frac{\Delta\lambda_{R,N}(1 + 3\sin^2\lambda_R)^{7/12}(1-\bar{\omega})}{|\gamma\bar{\omega} - 2\gamma\bar{\omega}^2 + 1||1 - \gamma\bar{\omega}|^{4/9}}, \tag{1}$$

with $B_w$ indicating the wave amplitude, $\omega_m$ the mean wave frequency, $\Delta\omega$ the frequency width, $\bar{\omega} = \omega_m/\Omega_{ce}$ the normalized frequency, $\Omega_{ce}$ and $\Omega_{ceq}$ the local and equatorial electron cyclotron frequency, $\gamma$ the relativistic factor, $p$ the electron momentum, $\epsilon_{meq} = \Omega_{pe}/\Omega_{ceq}\sqrt{\omega_m/\Omega_{ceq}}$ where $\Omega_{pe}$ is the plasma frequency, $T(\alpha_{eq})$ the bounce

period, $\lambda_R$ the latitude of resonance, and $\Delta\lambda_{R,N}$ the latitudinal range of resonance (see details in Artemyev et al. (2013)). The precipitating differential energy flux within the loss cone can be estimated as $x(E)J(E, \alpha_{LC})$, where

$$x(E) = 2\int_0^1 I_0(Z_0\tau)\tau d\tau/I_0(Z_0), \tag{2}$$

being the index of loss cone filling, $J(E, \alpha_{LC})$ is the electron differential energy flux near the loss cone, $I_0$ is the modified Bessel function with an argument $Z_0 \simeq \alpha_{LC}/\sqrt{\langle D_{\alpha_{eq}\alpha_{eq}}\rangle \cdot \tau_{loss}}$ (Kennel & Petschek, 1966), and $\tau_{loss}$ is assumed to be half of the bounce period.

With an energy distribution of precipitating electrons within 0.1–30 keV, we estimate the impact ionization rate altitude profile using the parameterization model developed by Fang et al. (2010), covering isotropic electron precipitation from 100 eV up to 1 MeV. This model, derived through fits to first-principle model results, allows efficient ionization computation for arbitrary energy spectra. Atmospheric density and scale height data were obtained from the NRLMSISE-00 model (Picone et al., 2002). We model dTEC resulting from whistler-induced electron precipitation by integrating ionization rates over altitude and time, adopting an 8-s integration period to align with the temporal resolution of THEMIS wave spectra data. Although our analysis does not concern equilibrium densities and omits recombination and convective effects, this has little impact because we focus on relative dTEC due to short-time precipitation. It takes nearly 60 s for the background ionosphere to relax to an equilibrium density solution for 10-keV precipitation and longer for lower energies (e.g., Kaeppler et al., 2022). Our estimated dTEC also closely match observed dTEC values, underscoring the effectiveness of our modeling approach despite its approximation.

## 3. Results

On 3 July 2013, from 15:06 to 16:36 UT, the THEMIS E spacecraft flew westward over the FAIR GPS receiver station, coming within ∼20 km relative to FAIR when mapped to 300 km altitude. The space-ground observations have a close spatial and temporal alignment, allowing us to link between magnetospheric and ionospheric processes along the field line. The event occurred at $L \sim 7$, outside the plasmapause of $L_{pp} \sim 5.4$ (based on THEMIS E densities near 17:00 UT), near the magnetic local time (MLT) of 4.5 hr, and during a geomagnetic quiet time with $Kp \sim 1$ and $AE \sim 200$ nT. Figure 2a illustrates the trajectories of THEMIS E and the ionosphere pierce points of GPS40, GPS43, and GPS60 near FAIR, mapped to 300 km altitude. The position of THEMIS E is mapped along the field line to the ionosphere using the Tsyganenko T96 model (Tsyganenko, 1995) but the GPS satellites are mapped using line of sight. Of these GPS satellites, the GPS43 pierce points, moving eastward, were nearest to both the FAIR and THEMIS E footprints, exhibiting close longitudinal alignment. A notable conjugacy, marked by the bright red segment from 15:37 to 16:11 UT, occurred when the footprints of THEMIS E and GPS43 pierce points were within ∼100 km to each other and FAIR (Figure 2j). In Supporting Information S1, we also present the configuration when the satellites and their pierce points are mapped to an altitude of 150 km. This adjustment does not significantly alter the geometry of our conjunction event, but it does slightly reduce the scale of the satellite footpaths near FAIR.

Figures 2b–2d present THEMIS observations of whistler-mode waves. The observed wave frequencies were in the whistler lower band, spanning ∼ 0.2–0.5$\Omega_{ce}$, with a mean frequency $\omega_m \sim 0.35\Omega_{ce}$, and $\Delta\omega \sim 0.15\Omega_{ce}$, where the electron cyclotron frequency $f_{ce} \sim \Omega_{ce}/2\pi \sim 2.15$ kHz. Figure 2d shows that whistler-mode wave amplitudes $B_w$ range from several pT to over 100 pT, measured at 8-s cadence (black curve) and smoothed with 2-min moving averages (red curve). Short-term oscillations in $B_w$ on the order of tens of seconds were observed atop more gradual variations of several minutes. We use smoothed or averaged $B_w$ to estimate electron precipitation. Although direct waveform data for resolving whistler-mode wave normals were absent, we can infer wave normals based on the measured whistler spectra properties of $E/cB \ll 1$ (see Supporting Information S1) as well as from previous statistical whistler observations in the nightside equatorial plasma sheet (Agapitov et al., 2013; Meredith et al., 2021; W. Li, Bortnik, Thorne, & Angelopoulos, 2011). The whistlers propagate quasi-parallel to the magnetic field, with an assumed Gaussian wave normal width of $\Delta\theta \sim 30°$ and a latitudinal distribution within $\pm 30°$.

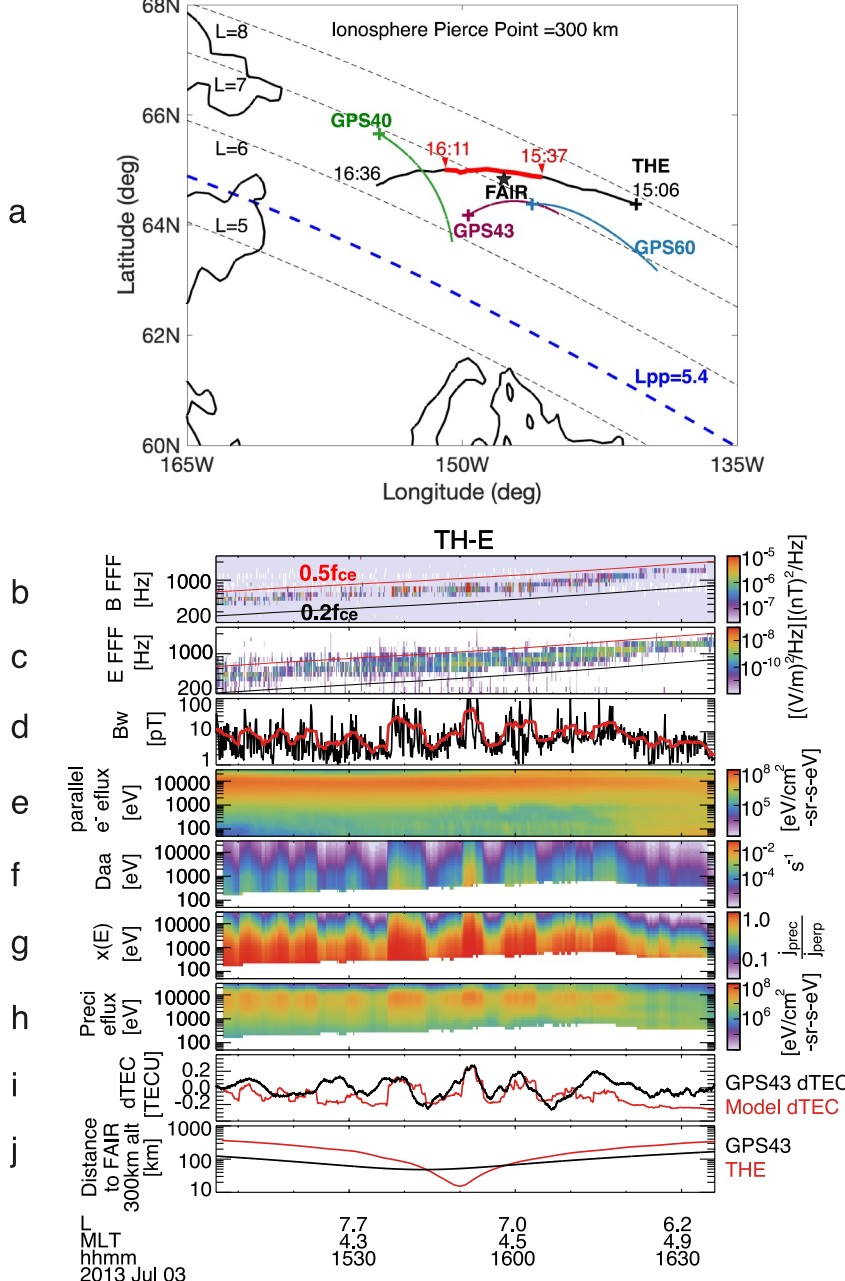

**Figure 2.** (a) Configuration of THEMIS E (black curve), GPS40, GPS43, and GPS60 satellites (green, purple, and blue curves), and the FAIR receiver (black star) in geographic coordinates, with THEMIS and GPS mapped onto 300 km altitude using T96 field tracing (THEMIS) or line of sight projection (GPS). The plus symbol indicates the start of the footpath. (b–e) THEMIS E magnetic field spectrogram, electric field spectrogram, whistler-mode wave amplitudes, and field-aligned (0°–22.5°) electron energy spectrogram. (f) Bounce-averaged electron diffusion rates. (g) Index of loss cone filling. (h) Whistler-driven precipitating electron energy spectrogram. (i) Comparison of whistler-driven model dTEC (red curve) and GPS43-observed dTEC (black curve). (j) Great-circle distances between THEMIS-E footpath (red curve) and GPS43 raypath (black curve) at IPP of 300 km relative to the FAIR station.

Figures 2e–2h display the measured plasma sheet field-aligned ($\alpha \sim [0°, 22.5°]$) electrons from 50 eV up to 25 keV, calculated diffusion rates $\langle D_{\alpha_{eq}\alpha_{eq}} \rangle$, estimated loss cone filling $x(E)$, and precipitating electron energy fluxes. Although $\langle D_{\alpha_{eq}\alpha_{eq}} \rangle$ and $x(E)$ increase at lower energies, the precipitating energy fluxes peak between 1 and 10 keV, exhibiting similar modulations as seen in the smoothed whistler-mode wave amplitude $B_w$. Electron

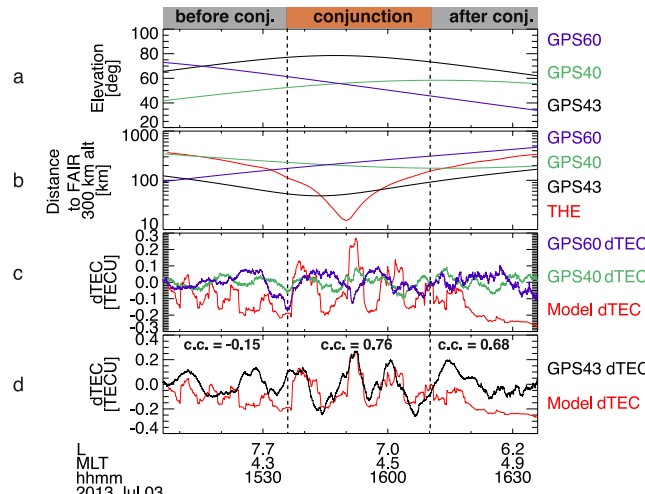

**Figure 3.** (a) Raypath elevation angles of GPS40 (green curve), GPS43 (black curve), and GPS60 (magenta curve). (b) Distances between THEMIS E footpath and GPS satellite pierce points relative to FAIR, displayed in the same format as Figure 1j. (c) Comparison between whistler-driven model dTEC and observed dTEC from GPS40 and GPS60, which were not in good conjunction with THEMIS or FAIR. (d) Comparison between whistler-driven model dTEC and GPS43-observed dTEC. The cross-correlation coefficients are −0.15, 0.76, and 0.68 during intervals before, during, and after conjunction, respectively.

precipitation fluxes below ∼200 eV are absent due to an energy threshold for electron cyclotron resonance interaction, with the lower limit primarily determined by the ratio $\Omega_{pe}/\Omega_{ce}$ (∼3 in our case).

Figure 2i compares modeled (red) and directly measured dTEC (black) from the GPS43 signal, revealing a nearly one-to-one phase correlation from 15:37 to 16:11 UT. This period of close correlation coincides with the conjunction of THEMIS E, GPS43, and FAIR, where their relative distances were within ∼100 km (Figure 2j). Outside this conjugacy period and further away from the FAIR station, the correlation decreases. Observed peak-to-peak amplitudes of dTEC reached ∼0.5 TECU, which is typical, though not extreme, for the nightside auroral region. This particular event occurred during quiet conditions; other events during storms may have much larger dTEC modulation amplitudes (e.g., Watson et al., 2015), though more challenging to have such reliable conjunction, especially given uncertainties in magnetic field mapping during storms (e.g., C.-L. Huang et al., 2008).

Figure 3 underscores the critical role of observation geometry and timing in detecting phase correlations between modeled and measured dTEC across three GPS satellites. Despite all three satellites having raypath elevation angles >40°—reducing the likelihood of multi-path effects (e.g., Jakowski et al., 1996)—only the GPS43 elevation reached 80° above the FAIR station zenith (Figure 3a). During the conjugacy period, the pierce points of GPS40 and GPS60 were distanced from FAIR by more than 200 km, while GPS43's pierce points remained within 100 km, coming within 20 km at its closest point (Figure 3b). Figures 3c and 3d reveal that the modeled dTEC (red curve) aligns poorly with GPS40 and GPS60 dTEC (blue and magenta curves), but a significant cross-correlation (∼0.8) emerges with GPS43 dTEC (black) during the conjugacy period. Before and after the conjunction, dTEC phase shifts reduce the cross-correlation to −0.15 and 0.68, respectively. Given the near-parallel longitudinal alignment of GPS43 pierce points and THEMIS E footprints (Figure 2a), the measured dTEC (black) potentially reflects both temporal and spatial/longitudinal modulations. These findings suggest that to reliably identify the electron precipitation responsible for dTEC requires precise spacecraft spatial alignment, optimal timing, and high raypath elevations.

The modulation of dTEC, electron precipitation, and whistler-mode wave amplitudes was linked to ULF wave activities in the Pc3-5 band (1.7–100 mHz). Figure 4a display the magnetic field perturbations measured by THEMIS E in the mean field-aligned coordinates, in which the parallel direction (∥, the compressional component) is determined by 15-min sliding averages of the magnetic field, the azimuthal direction (ϕ, the toroidal component) is along the cross product of $z$ and the spacecraft geocentric position vector, and the radial direction ($r$, the poloidal component) completes the triad. Magnetic perturbations are obtained by subtracting the 15-min mean field. During the conjunction, THEMIS E detected both compressional Pc5 waves (red curve) and poloidal Pc3-4 waves (blue curve). Figure 4b indicates that peaks in whistler-mode wave amplitudes approximately align with troughs of compressional ULF waves, with fine-scale whistler amplitudes primarily modulated by poloidal Pc3-4 waves (See Supporting Information S1). Strong Pc5 ULF waves were also recorded in the *H*-component magnetic field perturbations from magnetometers located at CMO and FYKN (Figures 4g and 4h), displaying a similar pattern but with greater amplitudes at FYKN, located slightly north of FAIR. The discrepancy between ground- and space-measured Pc5 waves potentially results from the localized nature of THEMIS-E observations (X. Shi et al., 2022) and the screening/modification effects of ULF waves traversing the ionosphere (Hughes & Southwood, 1976; Lysak, 1991; Lessard & Knudsen, 2001; X. Shi et al., 2018). Our observations imply that the ionospheric dTEC were linked to ULF-modulated whistler-mode waves and the associated electron precipitation (e.g., Coroniti & Kennel, 1970; W. Li, Thorne, et al., 2011; Xia et al., 2016; X. J. Zhang et al., 2020; L. Li et al., 2023).

Figures 4b and 4c compare small-scale/high-frequency fluctuations of whistler-mode wave amplitudes $B_w$ and dTEC, which was bandpass-filtered within the frequency range of 5–200 mHz. The small-scale dTEC fluctuations exhibit similar wave periods to $B_w$ fluctuations, evidently intensifying during the conjugacy period, yet lacking a

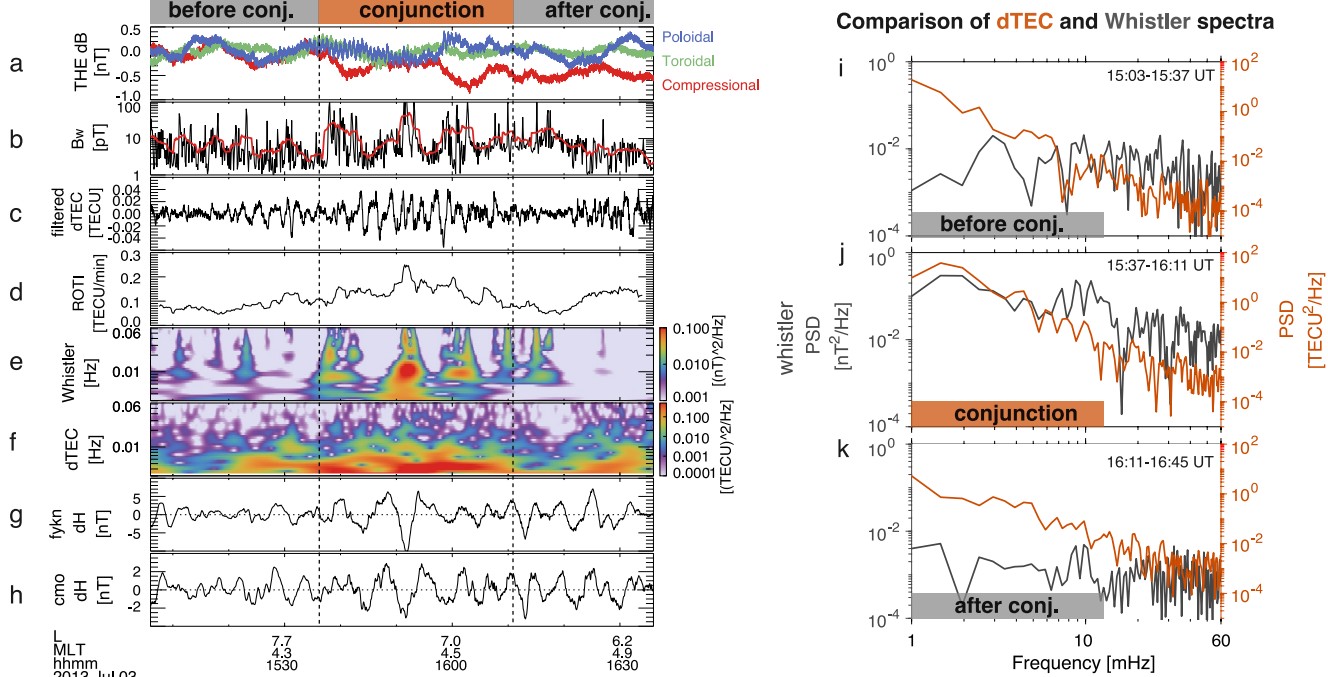

**Figure 4.** (a) THEMIS E magnetic field perturbations in the mean-field-aligned coordinates, exhibiting compressional- (red) and poloidal-mode (blue) variations. (b) THEMIS E whistler-mode wave amplitudes. The measured amplitudes are shown in black and smoothed in red. (c) dTEC bandpass filtered within 5–200 mHz. (d) *ROTI* from 200-s sliding window ensemble averaging. (e) Wavelet spectrogram of whistler-mode waves. (f) Wavelet spectrogram of GPS43 dTEC. (g) Ground-based magnetic field *H* component perturbations in 1.7–100 mHz from the Fort Yukon station. (h) Ground-based magnetic *H* component perturbations in 1.7–100 mHz from the College station. (i–k) Comparisons of dTEC (orange curves) and whistler-mode wave amplitude fluctuation spectra (gray curves) in 1–60 mHz measured before (k), during (j), and after (k) the conjugacy period.

clear phase correlation seen with larger scale perturbations in Figure 3d. Figure 4d shows the rate of TEC index (*ROTI*), that is, the standard deviation of the rate of TEC (*ROT*) (Pi et al., 1997), where $ROT = (dTEC(t + \tau) − dTEC(t))/\tau$ with $\tau = 10$ s, $ROTI = \sqrt{\langle ROT^2 \rangle − \langle ROT \rangle^2}$ using 200-s sliding averages. Significant increases in *ROTI* were observed within the region of whistler-driven TEC perturbations. However, in our case the GPS signal fluctuations were predominantly refractive, as negligible fluctuations were detected at frequencies above 0.1 Hz (McCaffrey & Jayachandran, 2017, 2019; Nishimura et al., 2023).

Figures 4e and 4f compare the wavelet spectrograms of whistler-mode wave $B_w$ and dTEC, displaying concurrent increases in wave power for both in the frequency range of ∼3 mHz up to tens of mHz. Figures 4i–4k present a more detailed amplitude spectra comparison before, during, and after conjunction. Notably, only during the conjunction, whistler-mode wave amplitudes and dTEC share similar power spectral density distributions in the 1–∼30 mHz range. The peaks in whistler spectra were slightly and consistently larger than those in dTEC spectra within 3–20 mHz by factors of 1.05–1.2 with an average of 1.15, aligning with expected Doppler shift effects on ionospheric TEC measurements. The Doppler shift results from relative motion of GPS raypath (with pierce point velocities of ∼46 m/s at 300 km altitude in our case) and propagating TEC structures (typically with velocities of several hundred m/s) (Watson, Jayachandran, & MacDougall, 2016): $f_{cor} = f_{obs}\left(1 + \frac{\mathbf{v}_{ipp} \cdot \mathbf{v}_{struct}}{|\mathbf{v}_{struct}|^2}\right)$, where $f_{cor}$ is the frequency corrected for relative motion. Watson, Jayachandran, and MacDougall (2016) found that 89% of their statistical events required a correction factor of 1.2 or less for the Doppler shift, consistent with our observations. The agreement between dTEC and whistler amplitude spectra supports that the observed dTEC resulted from electron precipitation induced by whistler-mode waves.

The average Doppler shift factor of ∼1.15 obtained from Figure 4j allows us to estimate the plasma drift velocity from $\vec{v}_{struct} \sim \vec{v}_{ipp}/0.15 \simeq 300$ m/s at the pierce point of 300 km altitude or 150 m/s at 150 km altitude. The spatial scales of the small-scale dTEC in Figure 4c can be estimated from $ds = \left(\left|\vec{v}_{struct}\right| − \left|\vec{v}_{ipp}\right|\right)dt$. The

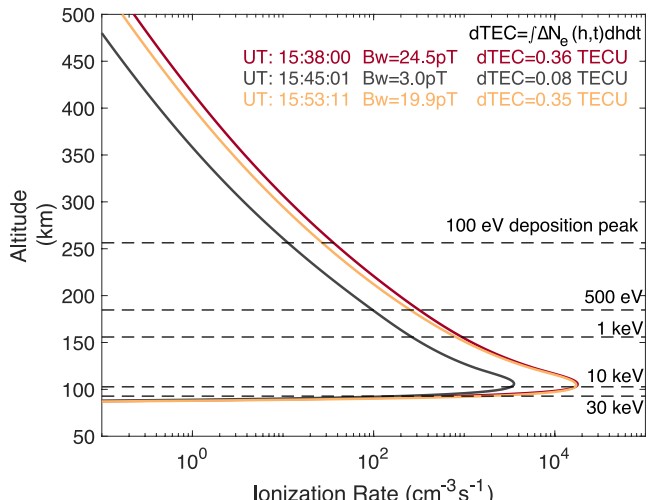

**Figure 5.** Ionization rate altitude profiles calculated at three time stamps of 15:38:00, 15:45:01, and 15:53:11 UT, corresponding to whistler-mode wave amplitudes of $B_w = 24.5$ pT (red curve), 3.0 pT (gray curve), and 19.9 pT (orange curve). The dTEC were calculated by integrating ionization rates over altitude and time (8s). The dashed lines mark the peak deposition altitudes of 100 eV, 500 eV, 1 keV, 10 keV, and 30 keV precipitating monoenergetic electrons.

resulting wavelengths are ∼10–30 km at the pierce point of 300 km altitude or ∼5–15 km at 150 km altitude. In contrast, the larger-scale dTEC shown in Figure 3d have wavelengths of ∼100 km at 300 km altitude or ∼50 km at 150 km altitude. When mapped to the magnetosphere, the small-scale dTEC modulations correspond to a magnetospheric source region of ∼150–700 km, while larger-scale dTEC modulations suggest a source region of ∼1,000–2,500 km. These scales align with prior observations of the transverse scale sizes of chorus elements and their source regions (Agapitov et al., 2017, 2018; Santolík et al., 2003) and also with the azimuthal wavelengths of high-m poloidal ULF waves (Yeoman et al., 2012; X. Shi et al., 2018; Zong et al., 2017).

Figure 5 indicates that the electron precipitation, induced by ULF-modulated whistler-mode waves, can cause significant increases in ionospheric ionization rate or column density, leading to dTEC of ∼0.36 TECU with a moderate whistler amplitude of $B_w \sim 25$ pT. Given that large-amplitude whistler-mode waves exceeding several hundred pT frequently occur in the inner magnetosphere (Cattell et al., 2008; Cully, Bonnell, & Ergun, 2008; Agapitov et al., 2014; Hartley et al., 2016; R. Shi et al., 2019), we anticipate even larger dTEC from such whistler activities. We defer a statistical study including storm time events and the potential connection with scintillation (e.g., McCaffrey & Jayachandran, 2019; Nishimura et al., 2023) for the future. In addition, the primary energy range of precipitation spans from ∼100 eV to ∼30 keV, contributing to density variations between ∼90–∼400 km (Berland et al., 2023; Fang et al., 2010; Katoh et al., 2023).

## 4. Discussion

Various mechanisms have been proposed that link ULF waves to dTEC and ionospheric disturbances in general (Pilipenko et al., 2014). Although dTEC might arise from direct ULF wave effects through convective and divergent flows, MHD Alfvén-mode waves do not directly alter plasma density. Furthermore, mode-converted compressional waves, if present due to Hall currents, are evanescent in the ionosphere (Lessard & Knudsen, 2001), resulting in negligible TEC perturbations (Pilipenko et al., 2014).

The vertical component of the $\vec{E} \times \vec{B}$ drift associated with ULF waves can induce vertical bulk motion of ionospheric plasma with a drift velocity $v_z = E_y \cos I / B_0$, where $I$ is the local magnetic inclination. This vertical transport can alter the altitude-dependent recombination rate, thereby contributing to electron density or TEC modulations (Pilipenko et al., 2014; Poole & Sutcliffe, 1987). These effects are potentially important in mid-latitude and equatorial regions (Yizengaw et al., 2018; Zou et al., 2017) but are expected to be less significant at high latitudes where the magnetic inclination is large. In our case, the magnetic inclination angle is such that $\cos I \sim 0.2$, and the magnetic perturbations are only a few nT, resulting in electric field perturbations $E_y < 1$ mV/m (Yizengaw et al., 2018). Based on similar estimations from Pilipenko et al. (2014), the resulting changes in $dn_e/n_e$ or dTEC/TEC are only 0.04%, corresponding to dTEC of <0.01 TECU given a background TEC of ∼20 TECU. This level of dTEC is insignificant compared with the observed 0.5 TECU. Moreover, the timescales of TEC changes due to recombination rate changes associated with vertical plasma motion are typically longer than 1 hr (Heelis et al., 2009; Maruyama et al., 2004; Yizengaw et al., 2006), which are much larger than the ULF modulation timescales of several minutes observed in our case. Therefore, the observed ULF-modulated high-latitude dTEC are unlikely to be explained by vertical plasma transport and recombination rate changes in the F region.

The periodic horizontal drift of ULF waves could produce noticeable TEC modulation across a horizontal density gradient, via the advection term $\vec{v} \cdot \nabla n_e$ (Pilipenko et al., 2014; Poole & Sutcliffe, 1987; Waters & Cox, 2009). This modulation may be enabled by a pre-existing east-west density gradient, which was suggested to produce TEC modulation of ∼2% with 5 nT magnetic perturbations near the terminator (Waters & Cox, 2009). However, our event was on the nightside, away from the terminator. The advection may arise from a latitudinal density

gradient coupled with ULF $\vec{E} \times \vec{B}$ drifts. Pilipenko et al. (2014) estimated that this latitudinal advection could contribute to dTEC/TEC of $\sim 0.1\%$ at auroral latitudes, corresponding to dTEC $\sim 0.02$ TECU in our case. In general, Poole and Sutcliffe (1987) theoretically derived the advection-induced TEC modulation as dTEC/TEC $\sim 2E_y/\omega B_0 L$, where $L$ is the horizontal gradient scale. If we take $E_y \sim 1$ mV/m, $\omega \sim 10^{-2}$ s$^{-1}$, $L \sim 30$ km, the resulting dTEC/TEC is only 0.17%. Thus, ULF-induced horizontal transport also cannot explain our observed dTEC modulation of $\sim 0.5$ TECU.

A non-linear "feedback instability" mechanism may modify ULF wave dynamics, causing field-aligned current striations and significant bottom-side ionospheric density cavities and gradients (Lysak, 1991; Streltsov & Lotko, 2008). Furthermore, in the presence of pre-existing larger-scale density gradients, ULF-induced plasma flows may result in gradient drift instabilities and density striations and irregularities with scale sizes less than $\sim$ 10 km (Basu et al., 1990; Gondarenko & Guzdar, 2004; Kelley, 2009; Keskinen & Ossakow, 1983; Nishimura et al., 2021; Spicher et al., 2015). Additionally, electron precipitation and Joule heating are important factors to consider in the auroral region (e.g., Deng & Ridley, 2007; Meng et al., 2022; Sheng et al., 2020).

Detecting one-to-one phase correlation between ground-based ULF waves and dTEC may be challenging, largely due to ionospheric screening effects on ULF waves (Hughes & Southwood, 1976), with only a few exceptions noted during storm times (Pilipenko et al., 2014; Wang et al., 2020). However, this correlation has been frequently observed with spacecraft measurements of ULF waves (Watson et al., 2015; Watson, Jayachandran, Singer, et al., 2016; Zhai et al., 2021), indicating that magnetospheric processes may play an important role in driving ionospheric dTEC. Our findings support that magnetospheric whistler-mode waves, modulated by ULF waves in the Pc3–5 band, are responsible for these periodic dTEC through associated electron precipitation.

These results enhance our understanding of dTEC modulation by ULF waves, a topic widely discussed in the literature (Pilipenko et al., 2014; Skone, 2009; Wang et al., 2020; Watson et al., 2015; Watson, Jayachandran, Singer, et al., 2016; Zhai et al., 2021), and facilitates the integration of effects of magnetospheric whistler-mode waves into auroral dTEC models. Statistical modeling of whistler-mode and ULF waves has been improving for several decades (e.g., Agapitov et al., 2013; Artemyev et al., 2016; Claudepierre et al., 2010; Hartinger et al., 2015, 2022, 2023; M. Hudson et al., 2004; Ma et al., 2020; McPherron, 1972; Sandhu et al., 2021; Shen et al., 2021; Southwood & Hughes, 1983; Takahashi & Anderson, 1992; Tsurutani & Smith, 1974; Tyler et al., 2019; W. Li, Bortnik, Thorne, & Angelopoulos, 2011; X. J. Zhang et al., 2020; Zong et al., 2017). Leveraging these wave effects and the associated electron precipitation can enhance physics-based modeling of ionospheric dTEC by providing better specifications of high-latitude drivers (Huba & Drob, 2017; Meng et al., 2016, 2020; Ridley et al., 2006; Schunk et al., 2004; Sheng et al., 2020; Verkhoglyadova et al., 2020; Zettergren & Snively, 2015). This wave-driven precipitation provides the dominant energy input to the ionosphere among all types of auroral precipitation (e.g., Newell et al., 2009), thus critically contributing to dTEC at high latitudes. As such, incorporating these magnetospheric phenomena is important for improving the accuracy of ionospheric dTEC models. This incorporation potentially benefits both GNSS-based applications and magnetosphere and ionosphere coupling science.

## 5. Conclusions

We present a detailed case study of ionospheric dTEC, using magnetically conjugate observations from the THEMIS spacecraft and the GPS receiver at Fairbanks, Alaska. This conjunction setup allows us to identify the magnetospheric driver of the observed dTEC. Our key findings are summarized below:

- Combining in-situ wave and electron observations and quasilinear theory, we have modeled the electron precipitation induced by observed whistler-mode waves and deduced ionospheric dTEC based on impact ionization prediction. The cross-correlation between our modeled and observed dTEC reached $\sim 0.8$ during the conjugacy period of $\sim 30$ min but decreased outside of it.
- Observed peak-to-peak dTEC amplitudes reached $\sim 0.5$ TECU, exhibiting modulations spanning scales of $\sim 5$–100 km. Within the modulated dTEC, enhancements in the rate of TEC index were measured to be $\sim 0.2$ TECU/min.
- The whistler-mode waves and dTEC modulations were linked to ULF waves in the Pc3-5 band, featuring concurrent compressional and poloidal mode fluctuations. The amplitude spectra of whistler-mode waves and dTEC also agreed from 1 mHz to tens of mHz during the conjugacy period but diverged outside of it.

Thus, our results provide direct evidence that ULF-modulated whistler-mode waves in the magnetosphere drive electron precipitation leading to ionospheric dTEC modulations. Our observations also indicate that to reliably identify the electron precipitation responsible for dTEC requires precise spacecraft spatial alignment, optimal timing, and high raypath elevations. Our findings elucidate the high-latitude dTEC generation from magnetospheric wave-induced precipitation, which has not been adequately addressed in physics-based TEC models. Consequently, theses results improve ionospheric dTEC prediction and enhance our understanding of magnetosphere-ionosphere coupling via ULF waves.

## Conflict of Interest

The authors declare no conflicts of interest relevant to this study.

## Data Availability Statement

THEMIS data are available at http://themis.ssl.berkeley.edu/data/themis/the/l2/. GPS RINEX data are publicly available from the NASA CDDIS archive of space geodesy data (https://cddis.nasa.gov/Data_and_Derived_Products/GNSS/high-rate_data.html). TEC data derived for this study is available at https://doi.org/10.48577/jpl.LGI5JS (Verkhoglyadova, 2024). The access and processing of THEMIS and ground-based magnetic field data from CMO and FYKN was done using SPEDAS V4.1, see Angelopoulos et al. (2019). The original CMO data are provided by the USGS Geomagnetism Program (http://geomag.usgs.gov) but can be accessed through http://themis.ssl.berkeley.edu/data/themis/thg/l2/mag/cmo/2013/. FYKN data are part of the Geophysical Institute Magnetometer Array operated by the Geophysical Institute, University of Alaska (https://www.gi.alaska.edu/monitors/magnetometer/archive). The ionosonde data from the Eielson station is available from https://giro.uml.edu/ionoweb/.

## Materials and Methods

Y. Shen acknowledges the use of the tool of ChatGPT4 to assist with text editing for some sentences of the Introduction and Results.

**Acknowledgments**

This work has been supported by NASA projects 80NSSC23K0413, 80NSSC24K0138, and 80NSSC23K1038. Portions of the research were carried out at the Jet Propulsion Laboratory, California Institute of Technology, under a contract with NASA. O.P.V., M.D.H., and X.S. were supported by NASA project 80NSSC21K1683. O.P.V would like to thank A.W. Moore (JPL) for data processing discussions. We are indebted to Emmanuel Masongsong for help with the schematic figure.

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
