## [Peer Review History · Agu Advances]

Peer Review History for 2024AV001255

[Version was not sent to review.]

Peer Review History for 2024AV001302

Reviewer #1

The authors conducted an event analysis of ULF-modulated whistler-mode waves in the magnetosphere, which drive electron precipitation, leading to ionization and Total Electron Content (TEC) changes. The in-situ satellite data and wave analysis presented in the manuscript are well executed and described. However, there is a lack of other supporting ionospheric observations, and potential competing processes in the ionosphere are not discussed.

The paper lacks supporting evidence from nearby ionosondes or radar, which could provide information about the ionospheric E and F region conditions during this event. Such information is crucial for selecting the appropriate Global Navigation Satellite System (GNSS) Ionospheric Pierce Point (IPP) height. Based on the ionization profile calculation in the paper, an IPP at 150 km or even lower seems more appropriate. This adjustment would reduce the scale of the satellite footpaths near FAIR but is more logical. If there is a clear F layer present during this event, the hmF2 information would be helpful in determining the IPP choice. The 450 km IPP selection is higher than the typical 350 km used in the Madrigal TEC database.

If there is an ionospheric F layer present during this event, the transport process of F-layer plasma should be considered and discussed to determine whether the TEC modulation is due to the transport process.

The conclusion that these phenomena are "...critically contributing to TEC perturbations at high latitudes. As such, incorporating these magnetospheric phenomena is vital for improving the accuracy of ionospheric TEC models." may be overstated, given that the TEC modulation shown in this event is only ~ 0.5 TECU, which is smaller than the typical TEC data uncertainty of $\sim 1-2$ TECU.

Reviewer #2

This is a review of "Magnetospheric control of ionospheric TEC perturbations via whistler-mode and ULF waves" by Shen et al.

General comments

Overall, the methods and data are relatively well described and the correlation between TEC changes, ULF waves, and whistler-mode waves is adequately presented. However, the manuscript is sometimes really hard to follow. While I think this study is sound, my main issue is the lack of clarity of some parts of the text and how the data/conclusions are presented to the reader.

There are too many acronyms one has to keep track of, I suggest reducing their number and focusing on those that are the most important. In addition, in the first half of the manuscript, the authors use many superfluous qualifiers that should be removed or given more specificity. For example, is it that important to emphasize the observations as "fortuitous"? Aren't a majority of events generally fortuitous unless you are running a specific campaign? What does this bring to the paper? If the authors want to say that these are rare events then they should mention it properly.

Throughout the text, very low frequency waves should not be hyphenated, same thing for ultra low frequency waves. I also think THEMIS E does not require hyphens but that might be just personal preference.

Related to the above as well, the ULF definition is given too many times: in the abstract, plain language abstract, introduction, and multiple times later on in the text. Please try to reduce this. Maybe just abstract and introduction? Same for TEC, dTEC, GPS etc. Maybe describing these acronyms once (or twice) should be enough.

Several links provided are not working. See Open Research comments below.

Detailed comments

** Introduction

48. How is fortuitous relevant here?

55. What does "physics-based" mean here?

82. If you will not use GPS receivers again, I would recommend not using the acronym. We have too many in the text. Same for TIDs if this is not going to be a big part of the manuscript.

110. Since you're using equations later, it might be best to use proper vectors and not just bold for $E \times B$ drift.

133. What does density inversion mean?

140. What postulation are you referring to?

164. Again, why is fortuitously used here? What do you want to say about the event?

** Figure 1. While I understand this is a representative diagram, I believe that putting your return field line from the poles to the equators might be misleading. Would it be possible to make it more accurate? Otherwise, this figure is quite nice.

** Data and Methodology

Please reduce the use of useless acronyms. If we are not going to see GIM, PRN, or DFB later in the text there is no need for the acronym. We have already so many acronyms to keep track of, adding more just makes the text overly cumbersome. If necessary it can be limited to the figure captions.

Even IPP while it comes back in the text might not be necessary, you can just reduce to "pierce points" later in the text should you want a shorter version.

178. processed at the Jet Propulsion ...

183. Is there a particular reason why you consider 450 km alt within 300 km of FAIR and 150 km alt within 100 km? Could you please explain your choices here or refer to papers should this be related to previous selection choices/studies?

187. Why not consider elevation angles less than 30 degrees, why are these choices made? Is it a limitation of the instrument or a limitation of the data?

189. Why are you focusing on wave periods smaller than 25 min?

229. please remove neglect

** Results

237 - 239. This sentence is superfluous. Please be more precise if you want to emphasize why this is important and unique.

242. IPPs have already been defined multiple times. I like to re-instate that this might work better without an acronym as we have difficulties following the sentences.

244. Field-line traced? Please rewrite this sentence.

259 - 265. Please reword this paragraph to make it clearer. As it stands it is difficult to understand. Maybe break it into multiple sentences?

279. Is 0.5 TECU significant? Why?

303. Do we really need the MFA acronym if it's only mentioned once?

319. "These observations" refer to the results of this paper or the previously mentioned discrepancy or references. Maybe be more specific.

** Figure 2

While it may be possible that this is due to the PDF format, this figure is too small and cramped. We can barely read and understand the panels. This showcases the most important result of this paper but we can barely make sense of the description. It's hard to understand what corresponds to the labels with energies and fluxes, etc. Please find a way for the information to be clearer and easy to identify.

**** Figure 4**

This figure is also too cramped and crowded on the panels. Any chance you can make it bigger or simplify the panel description (or at least make it clear).

**** Discussion**

378. Please rewrite this sentence. I'm not entirely sure what you mean here.

**** Conclusions**

Please remove the acronym definitions we've already seen them multiple times in previous sections.

**** Open Research**

All THEMIS links are not accessible or do not exist.

The DOI link for Verkhoglyadova 2024 does not exist.

****References**

OK.

Peer Review History for 2024AV001302R

Reviewer #1

The authors have done a very good job answering my previous questions and concerns. My only remaining question is about the possibility of the TEC variations due to horizontal transport. In the revised manuscript, vertical transport has been discussed but not horizontal transport. It would be useful to add a short paragraph discussing its possible role.

Reviewer #2

This is a review for Magnetospheric control of ionospheric TEC perturbations via whistler-mode and ULF waves by Shen et al.

I believe the authors have properly addressed the most important points raised by both reviewers. The supplemental information and additional explanations in the text significantly improve the paper. This manuscript is ready for publication after the following very minor corrections:

Lines 183 - 193: Please cut this sentence into two parts for clarity. Otherwise nice explanation.

Figure 2: While I note that the figure has been improved it is still significantly cramped. Would it be possible to remove some information that is not necessary, such as some of the scales (do we need every 10^n to understand the scale on the right-side axis? you can also better adapt the numbers on the left-hand axis). At minimum, consider a top-to-bottom presentation so the figures are more readable.

Line 386: Santolik not SantoliK

Peer Review History for 2024AV001302RR

Reviewer #1

[Version was not sent to review.]